# Label-Focused Inductive Bias over Latent Object Features in Visual Classification

**Ilmin Kang**[1]**, HyounYoung Bae**[2]**, Kangil Kim**[*]
AI Graduate School, GIST[†], Republic of Korea
{kangilmin0325, bonheur606060, kangilkim}@gmail.com

## Abstract

Most neural networks for classification primarily learn features differentiated by input-domain related information such as visual similarity of objects in an image. This input-domain focused inductive bias, while natural, can unintentionally conflict with unexpressed yet implicitly utilized relations over latent objects in human labeling, referred to *Undescribed world knowledge (UWK)*. Such conflicts can limit generalization of models by potential dominance of the input-domain focused bias in inference. To overcome this limitation without external resources, we introduce *Label-focused Latent-object Biasing (LLB)* training method that constructs label-focused inductive bias over latent objects determined by only labels as UWK. It has four steps: 1) it learns intermediate latent object features in an unsupervised manner; 2) it decouples their visual dependencies by assigning new independent embedding parameters; 3) it captures structured features optimized for the original classification task; and 4) it integrates the structured features with the original visual features for the final prediction. We implement the LLB on a vision transformer architecture, and achieved significant improvements on image classification benchmarks. This paper offers a straightforward and effective method to obtain and utilize undescribed world knowledge in classification tasks. The codes are available at https://github.com/GIST-IRR/LLB

## 1 Introduction

In many classification tasks, a neural network has a role of learning latent features from an input to determine its accurate output labels. In case of image classification, the model learns the features based on visual similarity and differentiation depending on their classes. This preference for the input-domain to handle the similarity of features is a common property in most well-known models such as convolutional neural network (Krizhevsky et al., 2017; He et al., 2016) and vision transformer (ViT) (Dosovitskiy et al., 2020) as their high sensitivity to input specific information (Park & Kim).

This behavior is naturally expected for neural networks, but the relations on the latent objects may be different in human labeling based on world knowledge. Figure 1 shows the clear example of the conflict in image classification. The figures on the left are visually very similar to those on the right, but the semantics of the labels are

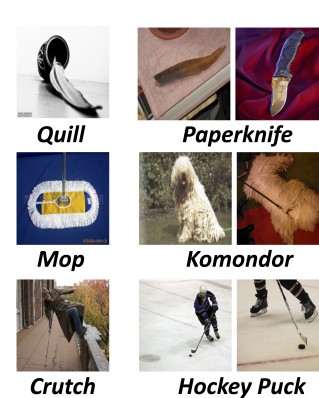

*Quill*     *Paperknife*

*Mop*     *Komondor*

*Crutch*     *Hockey Puck*

Figure 1: Visually similar but semantically different images when we see the relations of all objects that are not explicitly described in data

different in the real world. These samples may be regarded as just ambiguous and difficult samples for visual classification, but in fact we can differentiate them correctly by using unobserved relations in the data, such as the stick part of Mop or the ice ground for the Hockey puck.

---

[*]: corresponding author
[†]: Gwangju Institute of Science and Technology

This conflict between input-domain and potential knowledge on labels has been already discussed in various practical tasks, including visual question answering, cross-modal learning, and domain adaptation (Hendricks et al., 2018; Rohrbach et al., 2018; Zhou et al., 2019; Zhang et al., 2021; Cadene et al., 2019; Lemesle et al., 2022; Min et al., 2020; Radford et al., 2021) in various perspectives. However, these works have primarily focused on the inconsistency of rich information in observed data of two explicitly different domains (i.e. linguistic and visual relations of observed entities in each domain), while the conflict between input-domain information and undescribed relations over internal objects for determining output labels referred to *Undescribed World Knowledge (UWK)* in a single classification has been less investigated.

Such conflict in a single classification can limit generalization, because training and test data collected in the same labeling environment share the same UWK. If a model fails to learn such UWK, inference on test data can be dominated by input-domain focused inductive bias and lead to incorrect prediction. This limit of generalization by the conflict has not been discussed as an important issue to control so far. In visual classification, for example, the training data already contain a portion of UWK, as labeling can inform the category of internal objects in an image that are sufficiently abstract for the use as basic elements of the knowledge.

To this end, we propose *Label-focused Latent-object Biasing (LLB)* method for learning *label-focused inductive bias* over latent object features determined solely by categorization of labels, regarded as UWK. It has four sequential steps: 1) it learns intermediate latent object features in an unsupervised manner; 2) it decouples their visual dependencies by assigning new independent embedding parameters; 3) it captures structured features optimized for the original classification task; and 4) it integrates the structured features with the original visual features for final prediction. We empirically investigate the conflict and impact of LLB on latent feature distributions and implement it on ViT architecture. Our experiments on an image classification task show that LLB improves performance in both quantitative and qualitative analyses, demonstrating the benefits of regularizing the model without external resources.

Our contribution points are:

- We first raise the dominance of input-domain focused inductive bias of neural networks on inference that conflicts with undescribed world knowledge over latent objects by human labeling.
- We propose a training strategy, *Label-focused Latent-object Biasing (LLB)*, to obtain UWK and utilize it as label-focused inductive bias, and implement it on vision transformer for visual classification.
- We verify the proposed method in various image classification benchmarks with quantitative and qualitative analysis.

## 2 RELATED WORKS

**Conflict of Input Domain Focused Inductive Bias**  The conflict between input-domain and potential knowledge on labels has been discussed in various tasks. However, most works focus on the inconsistency of two different domain resources rather than the conflict of input-domain with undescribed world knowledge over latent objects in human labeling. In image captioning, (Hendricks et al., 2018) addresses the conflict of bias on visual contextual cues with gender-specific texts. (Rohrbach et al., 2018) raises over-reliance issue on language prior, leading to hallucination problem. (Zhou et al., 2019; Zhang et al., 2021) also observes that semantic inconsistency in the visual-text domain leads to hallucination, highlighting the issue of scarce aligned visual-text pairs. In visual question answering, (Cadene et al., 2019) introduces the impact of reducing dependencies on single-domain based statistical regularities when using both text and image input information. In cross-modality representation learning, vision networks leverage broader supervision of texts and adopts language bias in visual representation (Radford et al., 2021). However, cross-modal representations suffer inconsistency of language and visual domains (Pan et al., 2022; Lemesle et al., 2022). (Chen et al., 2019; Alberts et al., 2020; Li et al., 2020; Tan & Bansal, 2019) aim to make semantic aligned object representation using image-text paired dataset. To adapt CLIP (Radford et al., 2021) in a different domain, (Ma et al., 2022) made semantic alignments with large collection of semantic entities paired with images. (Pan et al., 2022) suggests semantic connection between differ-

ent modalities for semantic perceiving and addressed the limitation with external knowledge-graph based networks.

**Conflict Reduction via Knowledge Graph**   A traditional and direct solution of reducing the input-domain focused bias is to use external resources defined on labels as knowledge graphs. In visual understanding, (Marino et al., 2017) learns structured semantic representation over objects (Ren et al., 2015) with structured prior knowledge graph (Krishna et al., 2017; Miller, 1995). (Wang et al., 2018; Kampffmeyer et al., 2019) tried to inject knowledge graph and biases of language in the vision classifier to make zero-shot predictions. For open-world detection, (Yao et al., 2022) proposed large-scale unified concept dictionary with large-scaled image-text paired dataset to provide prior knowledge on detection. (Zhu et al., 2021) also learns dynamic knowledge graph over object class label from language domain bias to overcome the limitation of data scarcity in few-shot object detection. However, all these works focus on the effective use of external knowledge while UWK of the target training data has not been utilized and even recognized.

**Disconnecting Latent Objects from Inputs**   In visual understanding, (Marino et al., 2017) learns graph structures on internal objects, which are regarded as nodes, and integrates the features with external prior knowledge. To initialize each node feature, the object class label is used, which disconnects the visual dependency in the separate phase for graph neural networks. (Zhu et al., 2021) also learns dynamic semantic knowledge graph on object class label, which disconnects the visual dependency. However, visual disconnection in (Marino et al., 2017) are based on graph extraction and analysis. And (Zhu et al., 2021) may have learn separate embedding to input-domain, with the aim of representing language based semantic embedding over object class labels, rather than label-focused inductive bias. (Min et al., 2020) highlights the need for managing input-domain focused bias in regularizing representation for zero-shot classification, using decoupling effects by contrastive learning. Differently, our method focuses on conflicts in standard classification and direct extraction of UWK to induce decoupling and repositioning bias that is insensitive to input-domain features.

## 3 METHOD

### 3.1 PRELIMINARIES

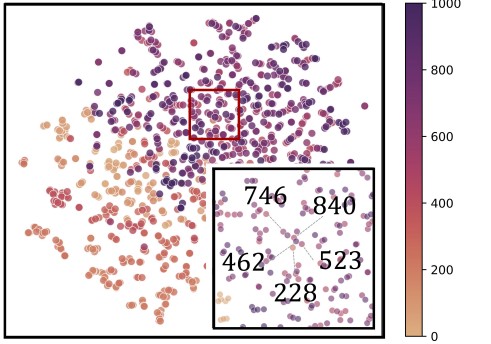

(a) Centroids of Class-wise Features (1000 classes)

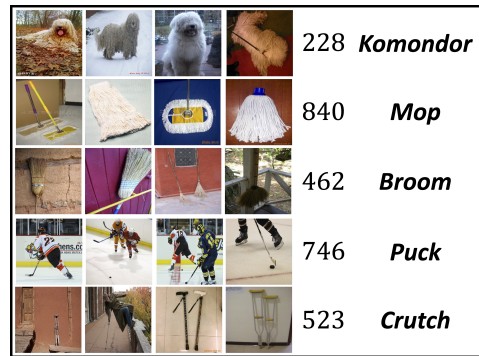

(b) Image Samples for Adjacent Classes

Figure 2: Input-domain focused inductive bias on features at the final layer, and its conflict with semantic difference on labels.

**Problem Confirmation in Class Distribution**   The problem to be addressed in this paper is the dominance of the input-domain focused bias on latent object features that conflicts with UWK. In Figure 2, the problem is clearly shown. The dots in the leftside figure represents the centroids of all features in each class of ImageNet, extracted from a MAE He et al. (2022) trained on the data. We zoomed in on an area where the centroids are closely located and selected almost adjacent five classes. However, in the rightside figure, their class labels are semantically unrelated. Although their visual similarity is sufficiently high, a human can correctly recognize the label. For example, the pole in a mop image may be confused with a tightened harness of a Komondor, but their shape

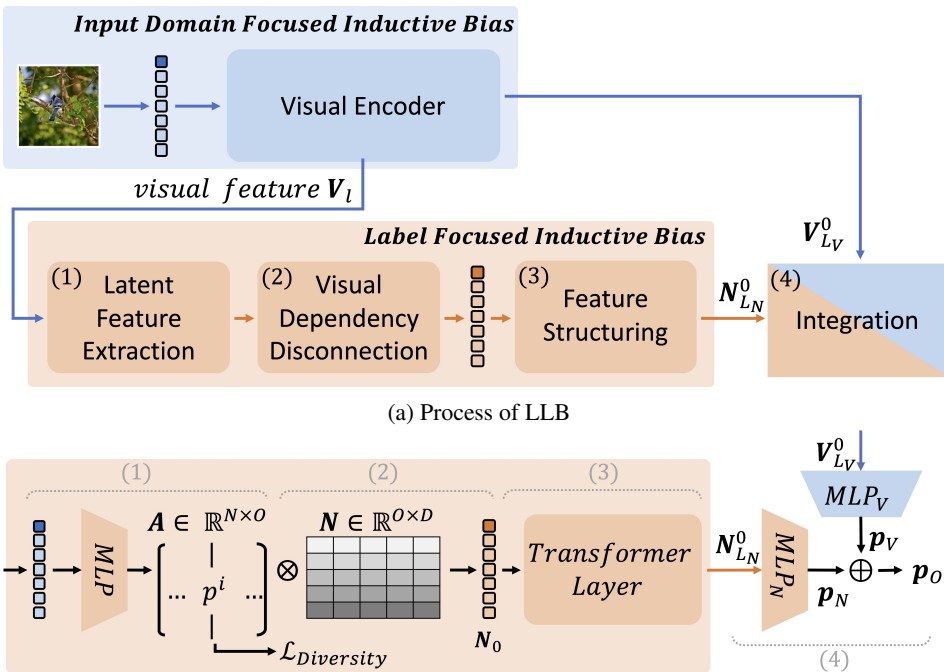

(a) Process of LLB

(b) LLB Module on Visual transformer

Figure 3: Overview of Label-focused Latent-object Biasing in image classification. (blue: visually determined, red: non-visually determined)

and angles associated with the other hairy features are distinguish the two classes. This observation shows an evidence for the dominance of the visual input-domain focused bias over the UWK. See Appendix C for additional problem confirmation using CNN and comparison with LLB.

**Motivation for Method: Label-based Grouping of Latent Objects Determines Their Similarity in UWK** Our motivation is that categorizing latent object features provides their similarity on the UWK without relying of external resources. A straightforward approach to utilize this *label-focused bias* is to train a classification model that takes the latent features as input. But, input-domain focused inductive bias may dominate over the label-focused bias in determining the feature positions. To address this issue, we propose training a separate plug-in classification model to capture the label-focused bias. Additionally, we propose disconnecting the direct forward path from an input to latent object features, which will then be fed into the model as new inputs.

**Notations for ViT** We start with brief recap of ViT (Dosovitskiy et al., 2020). ViT reshape input image $\mathbf{x} \in \mathbb{R}^{C \times H \times W}$ to patches $\mathbf{x}_p \in \mathbb{R}^{N \times (P^2 \cdot C)}$, where $(H, W)$ is original image's resolution with channel size $C$. Patches have size of $(P, P)$, and the derived total number $N$ of patches per images is $HW/P^2$. Patches are then projected into hidden dimension of size $D$ with additional positional embedding. Function $f(x)$, encodes patch representations with additional CLS token $\boldsymbol{c}_v$ by stacks of $L_V$ transformer (MHSA+FFNN) layers. We denote visual features as $\boldsymbol{V}_l = [\boldsymbol{v}_l^i]_{i=0}^N$, where $\mathbf{v}_l^i \in \mathbb{R}^D$ is hidden vector of $i^{th}$-patch of $l^{th}$-layer.

## 3.2 METHOD DESCRIPTION

**Overview of Architecture on Vision Transformer** We propose *Label-focused Latent-object Biasing (LLB)* method for visual classification problems as shown in Figure 3. Our network is based on a typical pre-trained ViT Dosovitskiy et al. (2020) that uses split image-patches as an input sequence, and is implemented in four sequential steps: 1) latent object extraction, 2) visual dependency disconnection, 3) object feature structuring, and 4) integration of non-visual and visual features for the final prediction. In the first step, the generated features from an intermediate $l$-th layer of the ViT (*visual features* $\boldsymbol{V}_l$) generates probability to select a latent object index among $O$ objects. In the next step for the disconnection, the most probable object index is mapped to separate learnable embed-

ding *non-visual features* $\boldsymbol{N} = [\boldsymbol{n}^i]_{i=0}^O \in \mathbb{R}^{O \times D}$ where $D$ is the equal dimension to visual features and $i$ indicates the object index. In the third step, the following transformer stacks (LLB layers) learn the structures of the non-visual embedding. At the end, the visual and non-visual features are integrated as an ensemble for the final decision.

**Latent Object Extraction from Visual Features**   The first step in visual object extraction involves quantizing visual features into a set of latent objects. This process is essential in limiting the number of non-visual features that may be associated with UWK, thereby ensuring that computational cost constraints are met. To implement the quantization, we pass a visual feature of each image patch into an `MLP`. This is then followed by a `Softmax` operation at a temperature of $T(= 0.1)$. The resulting vector, $\boldsymbol{p}^i$, determines the probability of selecting the latent object of index $i$. The following process is applied to all patches generated from an image.

$$\boldsymbol{p}^i \quad = \quad \texttt{Softmax}(\texttt{MLP}(\boldsymbol{v}_l^i)) \quad , \quad \boldsymbol{p}^i \in \mathbb{R}^O \tag{1}$$

The clarity of objects is influenced by their number, therefore we consider it a hyper-parameter. The effective range for this parameter is detailed in Table 7a.

To effectively represent groups of visual features using a limited number of latent objects, we incorporated a loss function that promotes diversity of probabilities, as described in  (Van Gansbeke et al., 2020; Mustafa et al., 2022).

$$\mathcal{L}_{Diversity} \quad = \quad -\mathcal{H}(\tilde{p}), \quad \text{where } \tilde{p} = \frac{1}{N} \sum_{i=0}^N \boldsymbol{p}^i \quad \text{and} \quad \mathcal{H}(p) = -\sum_{i=1}^O p^i \log(p^i) \tag{2}$$

where $\mathcal{L}_{Diversity}$ makes the averaged probability vectors of all objects uniform, encoursing even assignment across all objects. $\mathcal{L}_{Diversity}$ is combined with downstream task loss (e.g., cross-entropy loss) using the balancing parameter $\lambda(=0.5)$.

**Visual Dependency Disconnection**   The core idea of visual disconnection is to assign separate embedding parameters to visually determined latent objects. This process interrupts the gradient flow of the embedding parameters stemming from input-based differentiation, ensuring that subsequent training on the embedding is not overly influenced by this differentiation. To achieve this, we employ a straightforward *Disconnect* network, as illustrated in Figure 3b.

$$\boldsymbol{N}_0 \quad = \quad \texttt{Disconnect}(\boldsymbol{A}, \boldsymbol{N}) = \boldsymbol{A} \times \boldsymbol{N} \tag{3}$$
$$\boldsymbol{A} \quad : \quad \text{A matrix of patch-wise probability vectors to select latent objects}$$
$$\boldsymbol{N} \quad : \quad \text{A matrix of non-visual features in disconnected parameters from input}$$

The `Disconnect` network straightforwardly assigns disconnected embedding parameters to latent objects and subsequently produces patch-wise non-visual features, denoted as $\boldsymbol{N}_0 = [\boldsymbol{n}_0^i]_{i=0}^N$, used as inputs of LLB module. It operates matrix multiplication of assign matrix $\boldsymbol{A} = [\boldsymbol{p}^i]_{i=0}^N \in \mathbb{R}^{N \times O}$, composed of the generated probability vectors ($\boldsymbol{p}^i$s in Equation( 1)) to select latent objects, and the separate trainable non-visual feature $\boldsymbol{N}$. Rather than using the `argmax` function, matrix multiplication with the assign matrix is employed. This makes it differentiable while interrupting the input-based differentiation. It is worth noting that we utilize a separate CLS token, denoted as $\boldsymbol{n}_0^0$, for downstream tasks.

**Non-visual Feature Structuring**   Structuring non-visual features is the core step to redefining the similarity of features built over latent objects via solely the categorization of labels. The structuring method, referred to $g(\boldsymbol{X}, \boldsymbol{W})$, processes non-visual features $\boldsymbol{N}_0$ using weights $\boldsymbol{W}$ from structuring module to produce *structured non-visual features* $\boldsymbol{N}_{L_N}$. We implemented the function $g$ by a stacking $L_N$ layers of transformer (Vaswani et al., 2017) layer. We use transformer to extensively discern semantic relations within sequential input, as evidenced by works like  (Devlin et al., 2018; Radford et al., 2018; Park & Kim), owing to their flexible and scalable architecture. Adhering to the original method, we attached a CLS token to the head of non-visual feature sequence, denoted as $\boldsymbol{c}_n$, from the structured non-visual features for the final decision.

$$\boldsymbol{N}_{L_N} \quad = \quad g([\boldsymbol{c}_n || \boldsymbol{N}_0], \boldsymbol{W}) \tag{4}$$

where $\boldsymbol{N}_{L_N} = [\boldsymbol{n}_{L_N}^i]_{i=0}^N$. Compared to the original transformer layer, we exclude the positional embedding due to the weak dependence of clusters to specific positions in a sequence.

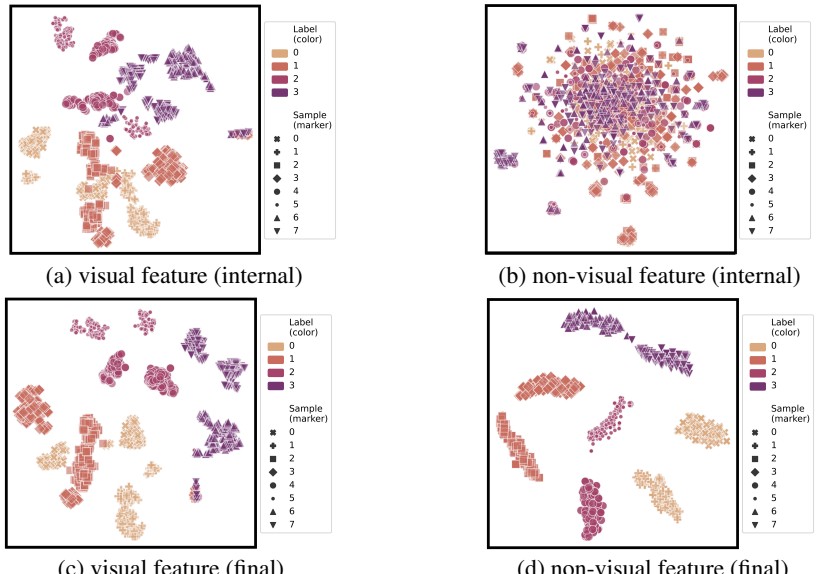

(a) visual feature (internal)    (b) non-visual feature (internal)

(c) visual feature (final)    (d) non-visual feature (final)

Figure 4: Comparison of object (image patch) feature distribution of each sample for four example classes at the end of training (Impact of visual disconnection: (a) and (b), Impact of structuring: (b) and (d), Impact of label-focused bias: (c) and (d)).

**Integration of Non-visual and Visual Feature** The primary objective of learning structured non-visual features is to achieve better generalization, as opposed to merely accurate prediction on observed data, which is effectively achieved by the original visual features. To leverage the strengths of both visual and non-visual features, LLB consolidates the output probabilities from both domains as described below.

$$\boldsymbol{p}_V = \texttt{SoftMax}(\texttt{MLP}_V(\boldsymbol{c}_v)) \tag{5}$$

$$\boldsymbol{p}_N = \texttt{SoftMax}(\texttt{MLP}_N(\boldsymbol{c}_n)) \tag{6}$$

$$\boldsymbol{p}_O = \alpha \times \boldsymbol{p}_V + (1 - \alpha) \times \boldsymbol{p}_N \tag{7}$$

To safely preserve the pre-trained probability for the visual features, we employ a separate classifier for predictions from non-visual features. Subsequently, we aggregate the output probability vectors, $\boldsymbol{p}_V$ and $\boldsymbol{p}_N$, from each classifier. A balancing parameter, $\alpha$, is introduced to modulate their respective contributions, as elaborated in Equation (7). The integrated probability $\boldsymbol{p}_O$ determines the final cross-entropy.

### 3.3 EMPIRICAL ANALYSIS ON FEATURE DISTRIBUTION

**Simple Settings** In this section, we provide empirical analysis of visual disconnection and structuring on feature representations. We extract them from a trained vanilla ViT (Dosovitskiy et al., 2020) described at Section 4. In Figure 4, each dot represents a feature corresponding to an image patch assigned to an latent object index, and the four figures illustrated via t-SNE (Van der Maaten & Hinton, 2008) show visual or non-visual feature distributions for the internal or final layer. Check Appendix Section B.1 for closer look at the figure.

**Impact of Visual Dependency Disconnection** Comparing Figure 4a with 4b, we observe that visual features for objects tend to cluster by both sample and class. In contrast, non-visual features exhibit a distribution closer to randomness. This disparity indicates the existence of the dominance of visual input-domain focused inductive bias.

**Impact of Non-Visual Feature Structuring** Figure 4b and 4d and show the change in the final features caused by the structuring non-visual features. The non-visual features are largely overlapped across samples and classes, showing their ambiguity for classification. However, the final features are more concentrated and separated by their class and sample. This implies that sufficient structuring is required for learning label-focused inductive bias useful in practical classification.

**Impact of Label-focused Bias** In comparison of Figure 4c with 4d, we can observe the difference of input-domain focused bias and label-focused bias. For most samples, the visual features are split into two clusters, with exceptions of sample 2 and 6. Notably, samples of different classes overlap in the cases of samples 2, 3, 4, 6, and 7. Furthermore, the sample area of two classes (0 and 1) are overlapped. In contrast, the label-focused inductive bias does not show any of the negative phenomena. This suggests that leveraging the feature concentration from the label-focused bias can further enhance generalization that was not achieved by input-domain focused inductive bias.

## 4 EXPERIMENT SETTING

Our LLB can be applied to any networks based on Transformer architectures. We evaluate the effectiveness of our non-visual feature by comparing it to vanilla networks in standard benchmarks. With pre-trained transformer networks, we first train our LLB in supervised manner, and measure its performance via an integration method (Section 3.2) that does not require any additional training. We also provide series of analyses of the impact of each component.

**Training Details.** Our LLB is built upon pre-trained classical visual feature backbones. We extract hidden vectors from backbone while keeping the backbone parameter frozen. For backbone, we use ViT (Dosovitskiy et al., 2020) networks. We use $l_V$-th layer and consider them as visual features $\boldsymbol{V}_{l_V} = [\boldsymbol{v}_{l_V}^0; \boldsymbol{v}_{l_V}^1; \cdots; \boldsymbol{v}_{l_V}^i]$. We found that, extraction from $l_V = L_V - 1$ layer showed best performance (Figure 7a in Appendix). LLB takes $\boldsymbol{V}_{l_V}$ and cluster them into $O$ latent objects. Based on our experiments (Figure 7b in Appendix), we use $O$=2048. We report the results of different $\alpha$ in Figure 7d in Appendix, and selected best one among them. Additional model settings are summarized in Table 3. We train our model with CrossEntropy loss with additional object diversity regularization term in Equation ( 2). Look at Table 4 in Appendix for detailed hyper-parameters we used. Our experiments are on $8\times$A100 with additional $4\times$A6000 GPUs for both reproduce baselines and training LLB.

**Image Classification.** We perform the evaluation on an image classification task. We show the effectiveness of the non-visual feature through the performance gained by adding our LLB to baselines. We use standard ImageNet (IN1K)(Deng et al., 2009), which consist of 1.28M training images with 1000 classes. We also use additional benchmarks including IN reassessed labels ImageNet-Real (IN-Real) (Beyer et al., 2020), scene recognition dataset Places356-Standard (Places) (López-Cifuentes et al., 2020), fine-grained and long-tailed iNaturalist2018 (iNat18) (Van Horn et al., 2018) dataset. For baselines, we first followed (Dosovitskiy et al., 2020; Steiner et al., 2021) to get vanilla ViT pre-trained using ImageNet21K (IN21K) (Ridnik et al., 2021). Also, to evaluate robustness of our method, we evaluate our method with diverse pre-training schemes. SWAG (Singh et al., 2022) is weakly-supervised ViT pre-trained via weakly supervision with hashtag labels (IG3.6B). Additionally, we use self-supervised method MAE (He et al., 2022) trained on IN1K. MAE is trained to reconstruct masked portion of an image. For some models, we used parameter weights from open source[1][2][3]. For others, we followed the training details described on (Singh et al., 2022; He et al., 2022; Singh et al., 2023) with 5 runs. All reproduced and official performance are reported in Table 1.

## 5 RESULT AND DISCUSSION

### 5.1 QUANTITATIVE ANALYSIS

**Performance** In Table 1, the performance before and after applying LLB is shown. Overall performance is significantly improved compared to the pre-trained models. Additionally, LLB remains effective even when applied to larger base models.

We also examine the robustness to different pre-training schemes. In these experiments, we replace supervised pre-trained ViT with weakly (SWAG) and self (MAE) supervised ViT. The results in Table 1 show that the different pre-training schemes benefit from label-focused inductive bias on

---

[1]ViT: https://github.com/huggingface/pytorch-image-models

[2]MAE: https://github.com/facebookresearch/mae

[3]SWAG: https://github.com/facebookresearch/SWAG

| Model | Pre. | Resolution | | Image Classification (Top1 acc.) | | | |
|---|---|---|---|---|---|---|---|
| | | Pre. | Fine. | IN1K | IN-Real | Places365 | iNat18 |
| ViT B/16* | IN1K | 224 | 224 | $79.00_{0.00}$ (77.91) | $83.76_{0.00}$ (83.57) | - | - |
| + LLB (Ours) | - | 224 | - | $79.43_{0.03}$ | $84.25_{0.02}$ | - | - |
| ViT B/16* | IN21K | 224 | 224 | $84.40_{0.00}$ (83.97) | $88.55_{0.00}$ (88.35) | - | - |
| + LLB (Ours) | - | 224 | - | $84.80_{0.01}$ | $88.90_{0.02}$ | | - |
| ViT L/16* | IN21K | 224 | 224 | $85.68_{0.00}$ (85.15) | $89.05_{0.00}$ (88.40) | - | - |
| + LLB (Ours) | - | 224 | - | $85.92_{0.02}$ | $89.26_{0.01}$ | - | - |
| MAE B/16† | IN1K | 224 | 224 | $83.63_{0.00}$ (83.60) | $88.29_{0.00}$ ( - ) | $57.84_{0.07}$(57.90) | $74.20_{0.05}$ (75.40) |
| + LLB (Ours) | - | 224 | - | $83.78_{0.02}$ | $88.40_{0.02}$ | $57.90_{0.06}$ | $74.32_{0.06}$ |
| MAE L/16† | IN1K | 224 | 224 | $86.08_{0.00}$ (85.90) | $89.63_{0.00}$ ( - ) | $59.60_{0.06}$ (59.40) | $80.06_{0.06}$ (80.10) |
| + LLB (Ours) | - | 224 | - | $86.12_{0.01}$ | $89.65_{0.02}$ | $59.70_{0.05}$ | $80.00_{0.06}$ |
| SWAG B/16‡ | IB3.6B | 224 | 384 | $85.28_{0.00}$ (85.30) | $89.00_{0.00}$ (89.10) | $58.90_{0.13}$ (59.10) | 79.78 ($79.90_{0.06}$) |
| + LLB (Ours) | - | 224 | - | $85.35_{0.04}$ | $89.10_{0.09}$ | $59.17_{0.02}$ | $79.86_{0.03}$ |

Table 1: Top-1 accuracy($Acc_{std}$%) on four image classification benchmarks (red: positive, blue: negative). We report the results from the paper in parentheses(*: the result of (Steiner et al., 2021), †: the result of (He et al., 2022), ‡: the result of (Singh et al., 2022)). The baseline results for IN1K and IN-Real are reproduced from the fixed open-source fine-trained models. For models that reproduced our-self, we state standard deviation as well. For both reproduce and proposal, We report the results of 5 runs.

most of the evaluation benchmarks. Note that LLB does not use additional pre-training with large data and regularization such as (Zhang et al., 2017; Yun et al., 2019; Szegedy et al., 2016).

**Ablation Study**  Table 2 reports ablation study results. We configure LLB with ViT B/16 trained on IN1K as the main model, and assess performance by excluding each attribute. First, we study the impact of the disconnecting visual dependency (Visual Disc.). We do not use the disconnection method while remaining the structuring method. Results show that disconnecting visual dependency significantly improves performance than maintaining it. Structuring visual features shows worse results than the baseline, indicating that LLB does not benefit from simply adding additional parameters. Then, we evaluate the impact of implementation component. We ablate the diversity loss for latent object extraction (Diversity), and we also set the transformer to not use positional encoding (w/o Pos.). The results show that each configuration is required to obtain the best performance. Finally, the performance without our integration module with visual features (Integration) shows the worst performance.

| Model | Visual Disc. | Diversity | w/o Pos. | Integration | IN1K (Top1-$Acc_{std}$%) |
|---|---|---|---|---|---|
| LLB (Ours) | ✓ | ✓ | ✓ | ✓ | $\mathbf{84.80_{0.01}}$ |
| | | ✓ | ✓ | ✓ | $84.25_{0.04}$ |
| | ✓ | | ✓ | ✓ | $84.74_{0.01}$ |
| | ✓ | ✓ | | ✓ | $84.77_{0.02}$ |
| | ✓ | ✓ | ✓ | | $82.58_{0.12}$ |
| Baseline | | | | | $84.40_{0.00}$ |

Table 2: Ablation study results of 5 runs with random seeds. We study the impact of visual dependency disconnection (Visual Disc.), diversity loss (Diversity), not using positional encoding (w/o Pos.), and integration module with visual features (Integration).

## 5.2 QUALITATIVE ANALYSIS

In this section, we provide an in-depth qualitative analysis on visual examples where LLB successfully predicts.

**Final Feature Distribution**  Figure 5 visualizes the final feature distribution generated from visual features (left) and non-visual features (right). In the left figure, we can find some visually similar features that are classified under semantically different labels. In the right figure, we trace the location of the sample (e.g. the screwdriver, quill, dough images with circle marker), but these samples are adjacent to the sample in the correct class. This outcome demonstrates that the label-focused inductive bias can effectively refine features toward their correct classes and rectify incorrect decisions made on visually ambiguous samples.

**Object Map on All Patches**  Figure 6 shows the examples of latent object indices mapped to each patch of an image via visual-feature based latent object extraction. We selected the examples that are

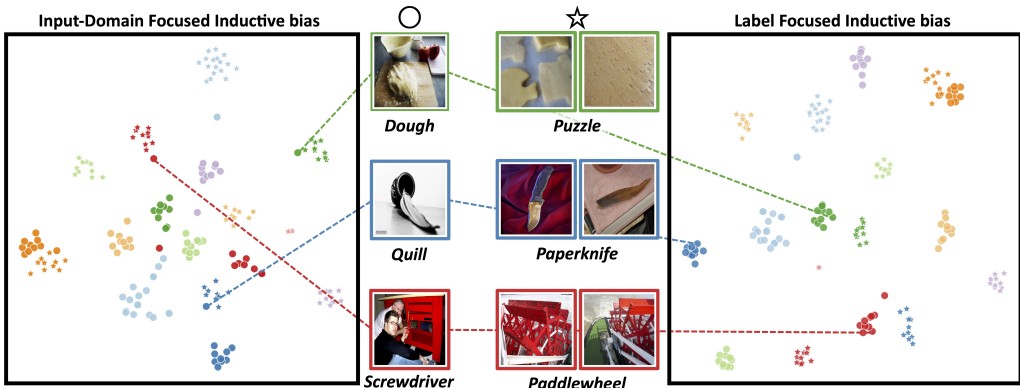

Figure 5: Final visual (*left*) and non-visual (*right*) feature for each sample. (color: a pair of two confused classes in visual feature based prediction, marker: one of the classes). Note that the circle sample is confused in visual feature map, but not in non-visual feature map.

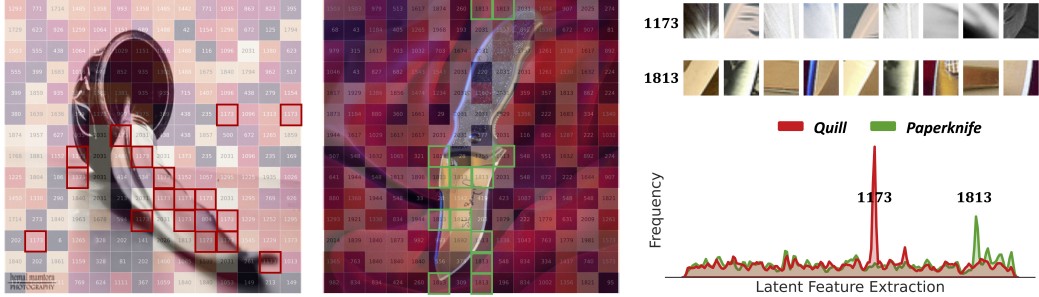

(a) Latent object map extracted from its overlapping image     (b) latent object frequency for each class

Figure 6: An example of extracted latent object distribution in a sample and over samples in each class. In (a), each tile shows an assigned latent object index of O(=2048) to an image patch. Red and green patches are dominating latent objects (1173 and 1813) in each image. (b) shows patch samples for the dominating objects and the frequency of the objects over all samples in each class.

visual-feature based models are confused. In Figure 6a, we observe the object 1173 over the feather and some background in the quill image, while the image patches on the paper knife are assigned to the different object 1813. As shown in Figure 6b, the two objects are distinct in the comparison of the other assigned image patches, and they are also identified as the most frequently used objects across all samples of the classes, serving as clear markers to distinguish between the classes. This finding suggests that latent objects derived from visual features inherently possess the ability to differentiate their respective classes. However, the input-domain focused inductive bias locates patches of the objects close together, confounding the prediction. In contrast, LLB disconnects the bias, and can leave the two objects as effective identifiers for correct classification. Additional object map results are in Figure 9 in Appendix.

## 6   CONCLUSION AND FUTURE WORK

**Conclusion**   In this paper, we highlighted the conflict of input-domain focused inductive bias and undescribed world knowledge over latent objects in human labeling. To advance regularization on this issue, we introduced *Label-focused Latent-object Biasing (LLB)* method that simply learns a separate classification model from intermediate object features disconnected from an input, and then integrates it with the original visual feature based classification model. Implementing the method on Vision transformer architecture, we can confirm its positive impact to model generalization through qualitative and quantitative analysis of its results in image classification tasks.

**Future Work**   Beyond using the plug-in model, effectively harmonizing the undescribed world knowledge with input-domain focused bias on a simple network still remains an open question. Also, we hope that our approach will evoke further research on diverse visual domains.

## 7 REPRODUCIBILITY STATEMENT

We demonstrate the reproducibility statement for this paper as follows.

- For reproducibility of our *Label-focused Latent-object Biasing (LLB)*, we first demonstrate our method in Section 3. Also we provide details of our implementation in Section 4 and Appendix A.1.
- To reproduce our implementation, we provide codes for training our LLB using various backbone models. Implemented codes are included in the supplementary material. We also provide README.md file for detailed description in the supplementary material.

## 8 ACKNOWLEDGMENTS

This work was supported by National Research Foundation of Korea (NRF) grant funded by the Korea government (MSIT) (No.2022R1A2C2012054, Development of AI for Canonicalized Expression of Trained Hypotheses by Resolving Ambiguity in Various Relation Levels of Representation Learning), Electronics and Telecommunications Research Institute(ETRI) grant funded by the Korean government (No.2011-2023-00046 Development of Large Korean Language Model Technology for Efficient Pre-training), if you intend to utilize the contents of this report, you must disclose that the research was funded by Electronics and Telecommunications Research Institute(ETRI), and Culture, Sports and Tourism R&D Program through the Korea Creative Content Agency grant funded by the Ministry of Culture, Sports and Tourism in 2022 (Development of service robot and contents supporting children's reading activities based on artificial intelligence No.R2022060001. contribution rate 25%) and appreciate the high-performance GPU computing support of HPC-AI Open Infrastructure via GIST SCENT.

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

# A APPENDIX

## A.1 IMPLEMENTATION DETAILS

**ViT architecture** Our LLB is built upon pre-trained ViT backbones. We use ViT-Base and it's scaled version ViT-Large for LLB. Table 3 demonstrates detailed information about model variants. We follow the settings from (Dosovitskiy et al., 2020) for ViT parameters.

LLB stacks $L_N$ layers of transformer layers to structure non-visual features. We report the impact of the number of layers on the LLB in Figure 7c, and selected values for $L_N$ based on the results. Our LLB adds additional `MLP` layers for latent feature extraction and stacks transformer layers for non-visual feature structuring.

| | ViT | | | | LLB | | | | | |
| Size | $L_V$ | $D$ | $FF$ | $H$ | $L_N$ | $D$ | $FF$ | $H$ | $l_V$ | $O$ |
|---|---|---|---|---|---|---|---|---|---|---|
| ViT-Base | 12 | 768 | 3072 | 12 | 6 | 768 | 3072 | 12 | 11 | 2048 |
| ViT-Large | 24 | 1024 | 4096 | 16 | 6 | 1024 | 4096 | 16 | 23 | 2048 |

Table 3: Details of model variants

**Hyper-parameter selection** Depending on the input-domain and UWK in a task, the conflict may be caused by different numbers of objects in different layers. So we set the layer to extract the objects and their number as hyper-parameter for tuning by tasks. We also set the number of layers to structure non-visual and the value of $\alpha$ for integration as a hyper-parameter, and measured their influence on the IN1K classification task. The effective range of the hyper-parameters are shown in Figure 7a.

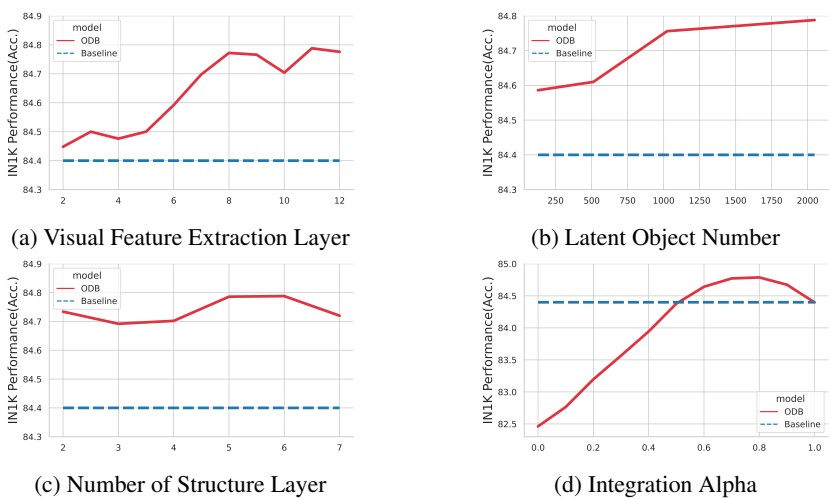

(a) Visual Feature Extraction Layer

(b) Latent Object Number

(c) Number of Structure Layer

(d) Integration Alpha

Figure 7: Impact of each hyper-parameter on IN1K image classification.

**Training details** We report our default training settings for IN1K image classification task in Table 4. For other evaluation benchmarks, only normalization values are changed. Table 5 reports the image classification performance on IN1K.

# B ADDITIONAL QUALITATIVE ANALYSIS RESULTS

**Object Clusters** Figure 8 shows successful examples of our latent object extraction. Each grid represents individual object cluster. We randomly sample clusters and clustered patches, and map them to the original image. For example, the second image in the first row has patterns like animal prints, and the second image in the second row has parts of fruit.

| Setting | Value |
|---|---|
| Epochs | 70 |
| Batch size | 1024 |
| Optimizer | Adam (Kingma & Ba, 2014) |
| Optimizer Momentum | $\beta_1 = 0.9, \beta_2 = 0.999$ |
| Learning rate: | |
|    Schedule | Cosine |
|    Peak | 1e-4 |
| Weight decay | 5e-4 |
| Loss | CrossEntropy |
| Augmentations: | |
|    Size | 224px or 384px |
|    RandAugment (Cubuk et al., 2020) | |
|      Magnitude | 9 |
|    Normalize | |
|      mean | [0.485, 0.456, 0.406] |
|      std | [0.229, 0.224, 0.225] |

Table 4: LLB training setting

| Model | Pre. | Params (M) | Resolution Pre. | Resolution Fine. | Top1 (acc.) IN1K |
|---|---|---|---|---|---|
| ViT B/16 | IN1K | 86.57 | 224 | 224 | 79.00 (77.91) |
| + LLB (Ours) | - | +46.45 | 224 | - | 79.43±.03 |
| ViT B/16 | IN21K | 86.57 | 224 | 224 | 84.40 (83.97) |
| + LLB (Ours) | - | +46.45 | 224 | - | 84.78±.01 |
| ViT L/16 | IN21K | 304.33 | 224 | 224 | 85.68 (85.15) |
| + LLB (Ours) | - | +80.80 | 224 | - | 85.92±.02 |
| MAE B/16 | IN1K | 86.37 | 224 | 224 | 83.63 (83.60) |
| + LLB (Ours) | - | +45.92 | 224 | - | 83.78±.02 |
| MAE L/16 | IN1K | 304.33 | 224 | 224 | 86.08 (85.90) |
| + LLB (Ours) | - | +80.80 | 224 | - | 86.12±.01 |
| SWAG B/16 | IB3.6B | 86.37 | 224 | 384 | 85.28 (85.30) |
| + LLB (Ours) | - | +45.92 | 224 | - | 85.35±.04 |

Table 5: Detailed top-1 accuracy on IN1K (accuracy in parenthesis: reference performance, red: positive, blue: negative).

**Object Map on All Patches with Other Images**   Figure 9 shows additional examples of object indices mapped to each patch of an image. In the mapped image in the *top* row, we found that the patches of the screwdriver are mapped to object 391 and the patches of the metal body are mapped to object 1736. From the frequency results on the right side, we can see that both features are distinctive features for each class.

## B.1   EMPIRICAL ANALYSIS RESULTS

We provide larger version of the visualization in Section 3.3.

## C   ADDITIONAL PROBLEM CONFIRMATION AND COMPARISON WITH LLB

Figure 11a from clearly shows the problem of the dominance of the visual-domain focused bias over the undescribed world knowledge over latent object in human labeling. The dots in the leftside figure represent the centroids of all features in each class of ImageNet, extracted from the ViT network trained on the data. When we zoomed in on a region of closed centroids, we found five adjacent but semantically unrelated class labels, shown on the rightside.

We also confirm this problem with the Convolutional Neural Network (CNN). We follow the same procedure that is described in Section 3.1, but replace ViT with the well-known CNN network ResNet50 (Krizhevsky et al., 2017). We used two versions of ResNet50 pre-trained with ImageNet training data. First, we used the pre-trained ResNet50 (Krizhevsky et al., 2017) in a supervised manner. For supervised pre-trained ResNet50, we followed the details of (Krizhevsky et al., 2017)

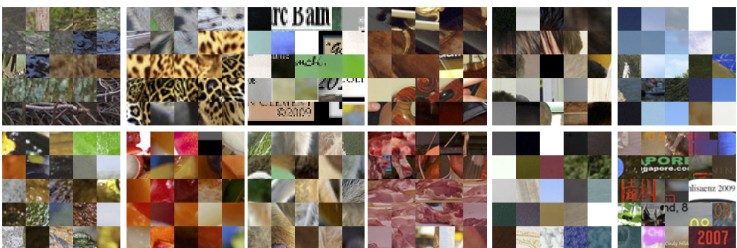

Figure 8: Positive examples of object latent clusters.

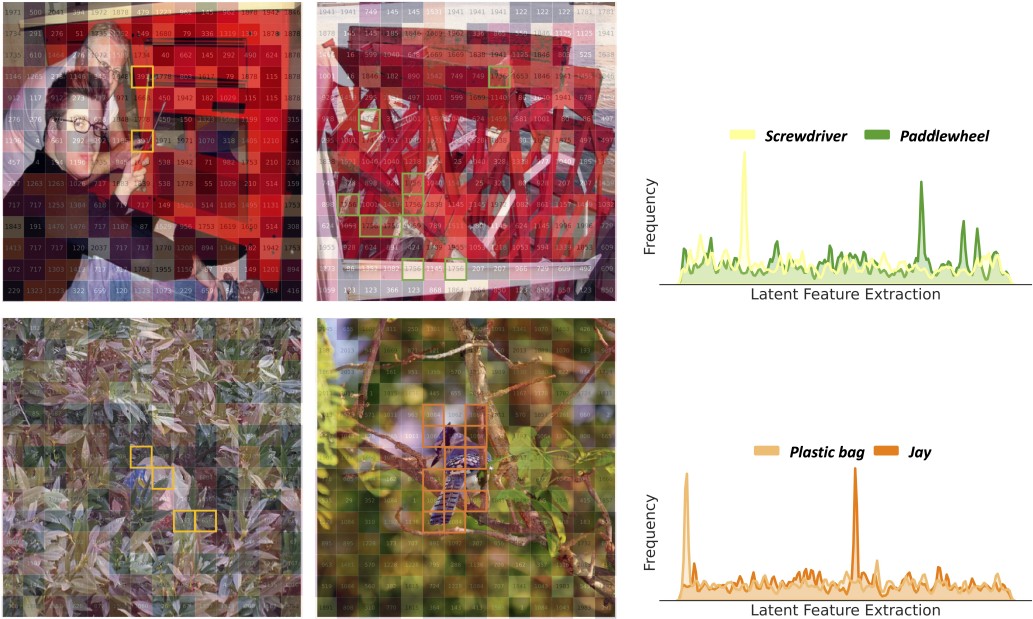

Figure 9: Additional example of object map on patches. In *left*, each tile shows an assigned object index to an image patch. *right* shows patch samples for the dominating objects and the frequency of the objects over all samples in each class.

and used parameters from open source[4] to reproduce a top-1 accuracy of 75.86% for IN1K (the reported performance from the open source is 76.13%). We also use ResNet50 pre-trained with self-supervised contrastive learning framework (Hadsell et al., 2006; Oord et al., 2018). Momentum Contrast (MoCo) (He et al., 2020) interpreted contrastive learning as dictionary look-up and built dynamic dictionaries with momentum-based moving average updates. MoCo v2 (Chen et al., 2020b) improved MoCo with the successes in (Chen et al., 2020a). We collected pre-trained ResNet50 weights using MoCo v2 from open source[5]. We then fine-tuned it using IN1K with the details described in (Chen et al., 2020b), and reproduced 77.01% top-1 accuracy in IN1K

## C.1 INPUT-DOMAIN FOCUSED BIAS IN CNN

Figure 11c shows the results of CNN in the classification benchmarks. In comparison with the bias in ViT 11a, semantically distinct classes ('840: Mop', '462: Broom', '764: Puck', and '523: Crutch') are still closely located, which is the common input-focused inductive bias of the dataset. This observation is an evidence for the conflict of the input-domain focused bias even in CNN.

---

[4]ResNet50: https://pytorch.org/vision/main/models/generated/torchvision.models.resnet50.html
[5]MoCo v2: https://github.com/facebookresearch/moco/tree/main

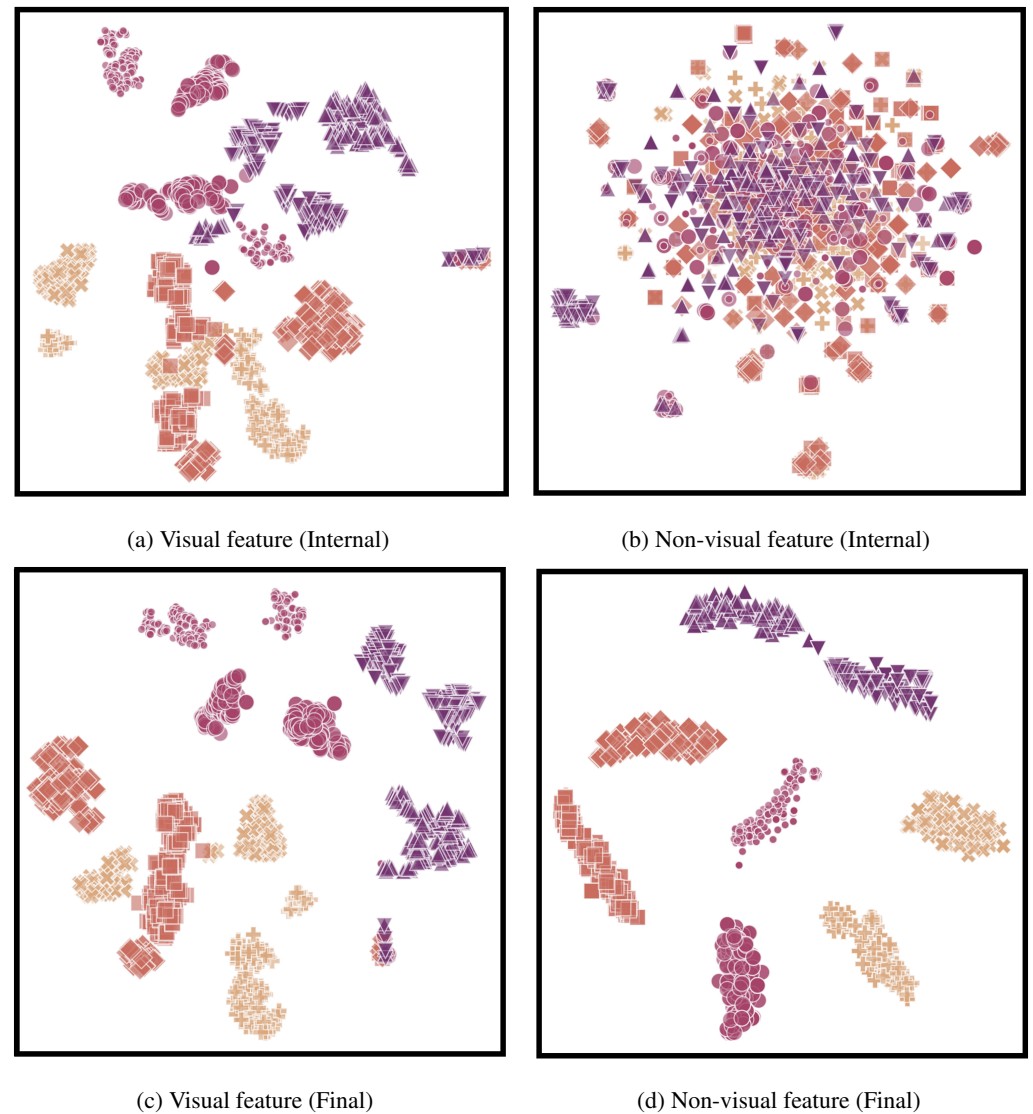

(a) Visual feature (Internal)

(b) Non-visual feature (Internal)

(c) Visual feature (Final)

(d) Non-visual feature (Final)

Figure 10: Feature distribution results.

## C.2 Input-Domain focused Bias in CNN with Contrastive Learning

Figure 11d shows the results of the CNN trained with contrastive learning. Using contrastive learning, the centroids of some classes (e.g. '523:Crutch' against '840: Mop', '462: Broom', '764: Puck') are slightly decoupled compared to supervised learning. However, this approach still fails to widen the gap between '462: Broom' and '746: Puck', where two class labels are visually similar in stick parts, but semantically distinguished by other objects. This observation shows that the input-domain focused bias is still strongly used in determining the features.

## C.3 Comparison with Label-focused Latent-object Biasing

Figure 11b shows the results of LLB using the same classes in Figure 11e. Compared to ViT (Figure 11a), where the centroids of all features of five classes are closed located, LLB shows distant gaps between classes. Also, while other networks fail to widen the gap between '462: Broom' and '746: Puck', LLB placed them in a distant location.

Additionally, we can see that '840: Mop' and '462: Broom' are closed located in LLB. We hypothesize that, the way of structuring over components of mop and broom are similar, making LLB to generate their features in a close location. In contrast, the other methods placed '840: Mop' and '462: Broom' in relatively more distant locations. This observation implies that LLB can diminish the dominance of the visual input-domain focused bias, and introduce a distinct bias, considered as the label-focused inductive bias.

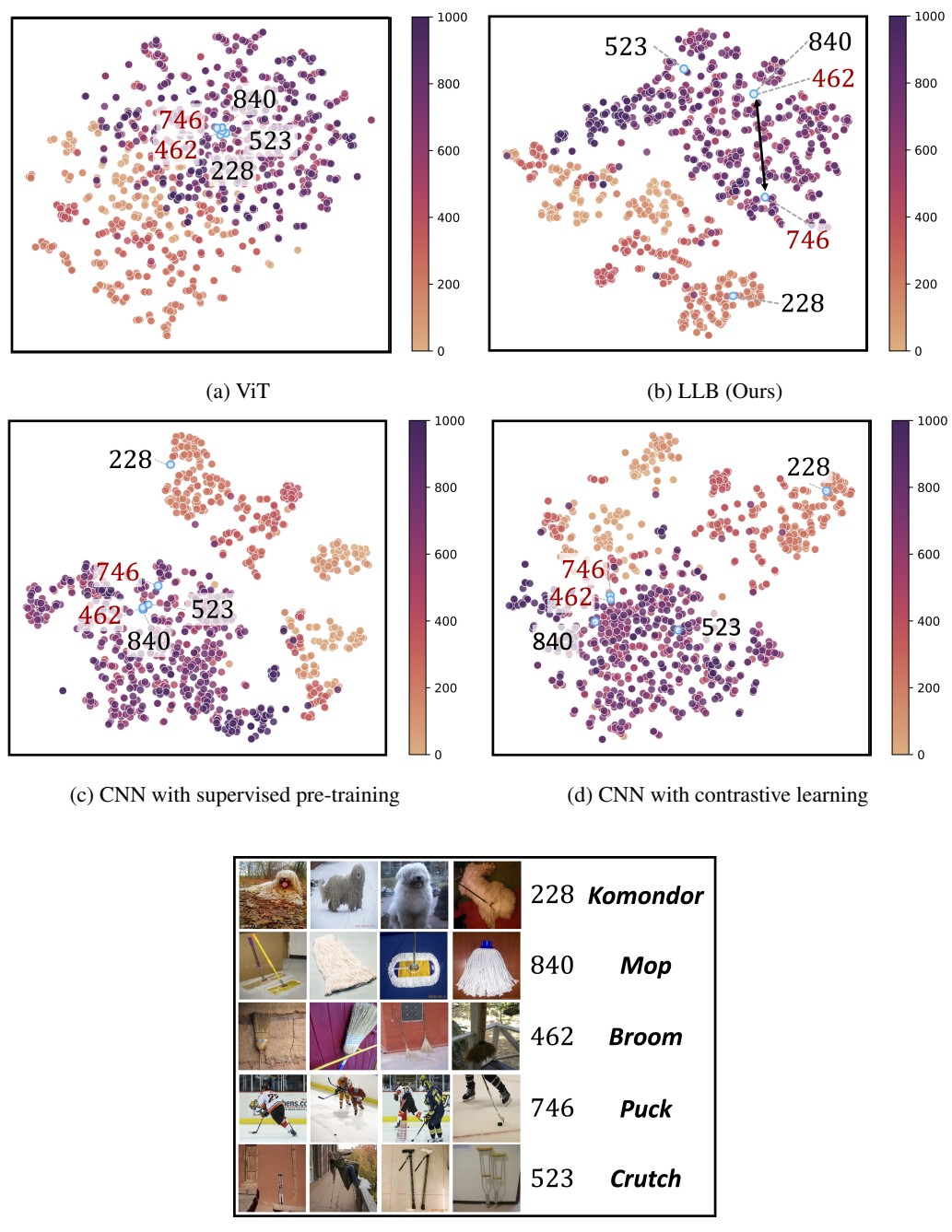

Figure 11: Comparison of the distribution of centroids of all features in each class of ImageNet. Centroids of all output features from ViT: (a), LLB (Ours): (b), CNN with supervised pre-training: (c), CNN with contrastive learning: (d). We highlighted the dots of five classes in (e) with sky-blue color.

