# OpenReview forum: "Label-Focused Inductive Bias over Latent Object Features in Visual Classification"
_ICLR.cc/2024/Conference — ICLR 2024 poster_

### Official Review · Reviewer_7b4C · 2023-10-30

**Soundness:** 2 fair
**Presentation:** 3 good
**Contribution:** 2 fair
**Rating:** 6
**Confidence:** 3

**Summary:**

The paper shows that neural networks tend to focus on input-domain-related information, such as visual similarities, which can conflict with unseen relations determined by human labeling in the output domain. This conflict limits the generalization of models. To address this problem, the authors propose a training strategy called Output-Domain focused Biasing (ODB), which emphasizes inductive biases based on output labels. ODB consists of four steps: learning intermediate latent object features, decoupling visual dependencies, capturing structured features optimized for classification, and integrating these features for prediction.

**Strengths:**

1. The paper aims to create inductive biases based on output labels, in order to avoid the dominance of input-domain focused bias. The motivation is relatively novel.
2. The paper is well-structured and written in an accessible manner.

**Weaknesses:**

1. The experimental improvements over the baselines are relatively modest, suggesting that the significance of Output-Domain focused Biasing (ODB) may be limited.

**Questions:**

Is it possible for the authors to validate the efficacy of ODB in the context of domain adaptation? Is there a meaningful application of ODB in addressing input-domain bias when dealing with diverse visual domains?

---

> ### Author Response · Authors · 2023-11-17
> **Thanks for your comments!**
>
> Thank you for all your efforts and detailed comments.
>
> > Weakness1. The experimental improvements over the baselines are relatively modest, suggesting that the significance of Output-Domain focused Biasing (ODB) may be limited.
>
> Our results show consistent and statistically significant improvement. We hope that the contribution on raising a new problem and its generality are more considered rather than the significance of performance gain determined by an unclear threshold.
>
> > Question1. Is it possible for the authors to validate the efficacy of ODB in the context of domain adaptation?
>
> We can validate the efficacy in domain adaptation, but it requires more study, because domain adaptation allows not only tasks using the shared world knowledge, but also task-specific knowledge.
> We hope that the knowledge of ODB is sufficiently general to work in even target domains for adaptation, but the conflict of task-specific knowledge may dominate the validation results. Unfortunately, recognizing which knowledge causes the conflict in neural networks is still challenging as far as we know.
>
> > Question2. Is there a meaningful application of ODB in addressing input-domain bias when dealing with diverse visual domains?
>
> We think that potential applications of ODB are most supervised learning in visual domains where labeling may rely on some implicit knowledge of humans. Particularly, if a domain uses an input containing many latent objects to determine its label, it can be a more suitable application of ODB.

---

### Official Review · Reviewer_CuBT · 2023-10-31

**Soundness:** 4 excellent
**Presentation:** 3 good
**Contribution:** 3 good
**Rating:** 8
**Confidence:** 4

**Summary:**

This paper addresses the issue where vision neural networks learn latent features which are (naturally) too focused on the exact pixels that are part of the training dataset -- since this is all they see, ignorant of the real-world relationships between objects (as illustrated by how ignorant a neural network can be of the distance between a mop and a komondor dog). To remedy this, the authors propose a novel method (ODB) where an auxiliary loss is introduced in order to enforce diversity in the latent vectors across classes. This loss term is disconnected from the regular discriminative visual features, so as to not pollute it. The authors show across 5 random seeds that they achieve better results on ImageNet1K than well-known baselines.

**Strengths:**

* The ODB approach does not use additional pre-training with large datasets and/or regularization such as MixUp or CutMix
* The performance of the approach is inspected thoroughly both quantitatively and qualitatively.
* The authors include transparent results on how they selected their hyperparameters in the appendix.

**Weaknesses:**

* The term of output-domain knowledge is very vague/difficult to interpret. At one point in the introduction you instead write 'undescribed world knowledge', which I think refers to the same thing, and is more clear to me. Would it make sense to use another name than ODB? Specifically to change the OD to something else. This would make the paper clearer. Output domain is the domain where you test, which I think does not really do the job here.

* In the related work, a knowledge graph is mentioned briefly while describing the method of one work. However, I miss a small section about the general idea of including hierarchical information or knowledge-graph representations into visual representations, which these authors are not the first to explore. For that reason, it becomes a bit difficult to buy when the authors claim they are the first to raise the issue of the implicit 'output-domain' knowledge missing during training, and a more complete related work section here would make the paper stronger (see for example the related work of Pan et al.)

* The conclusion and future work section does not really contain recommendations for future work.

**Questions:**

### Detailed comments

* In A.1, you refer to Tables 7a and 7c although I believe you mean Figure 7 and 7c.
* Conclusion: the phrasing "harbors unseen knowledge from human labeling" was difficult to parse and ambiguous.
* Conclusion: "on vision transformer architecture" >> "on the Vision transformer architecture"
* Conclusion: "in qualitative and quantitative analysis on its results" >> "through qualitative and quantitative analysis of its results"
* Table 2: Why not show the standard deviation if in effect these results were run using 5 random seeds? It seems relevant in your comparison to the baseline which is quite close (84.80 vs. 84.40). You cannot use the term 'significant' in the Ablation study section of Section 5.1 if you do not show these standard deviations.
* Table 2: explain what w/o Pos. stands for in table caption, even if you also say it in the text (if somebody glances quickly at the table.)
* Formatting of references generally needed, use {} around text which should be case-sensitive in the .bib-file. (e.g., "mixup" >> "MixUp" for Zhang et al.)
* In the end of Section 5.2, it would be great to specify either in the text or in the figure caption which classes numbers 1173 and 1813 correspond to. Are they the quill and paperknife or other visually similar classes?
* 5.1 title: quantiTative*
* Section 4: "doesn’t" >> "does not" (too informal)
* Section 4: "provide series of analysis the impact" >> "provide a series of analysEs OF the impact"
* The second paragraph of the related work contains a duplicate sentence "Lemesle et al...". Should be removed.
* Fig. 3 is a nice figure. Howeve,r it currently says "L_diveristy"  whereas I think you want to say "L_diversity".

---

> ### Author Response · Authors · 2023-11-17
> **Thanks for your comments!**
>
> Thank you for all your efforts and constructive comments. We have highlighted the revisions in our paper with purple text color based on the recommendation.
>
> > Weakness1. The term of output-domain knowledge is very vague/difficult to interpret. At one point in the introduction you instead write 'undescribed world knowledge', which I think refers to the same thing, and is more clear to me. Would it make sense to use another name than ODB? Specifically to change the OD to something else. This would make the paper clearer. Output domain is the domain where you test, which I think does not really do the job here.
>
> We agree that the term  ‘output-domain knowledge’ does not intuitively align with our paper's interest. We revised terminology to avoid using 'OD' as follows
>
> Label-focused inductive bias is the inductive bias determined by categorization of latent objects based on only labels, while undescribed world knowledge indicates all unexpressed world knowledge on the objects in human labeling. We use the term ‘UWK’ for ‘Undescribed World Knowledge over latent objects in human labeling’
>
> We revised related sentences (including titles) to use the terminologies, too.
> 1)  ‘Output-domain focused inductive biasing’ (the method) -> Label-focused Latent-object Biasing (LLB)’
> 2) ‘output-domain focused inductive bias’ -> label-focused inductive bias
> 3) implicit relations over latent objects used for  human labeling(output-domain) -> undescribed world knowledge
>
> > Weakness2. In the related work, a knowledge graph is mentioned briefly while describing the method of one work. However, I miss a small section about the general idea of including hierarchical information or knowledge-graph representations into visual representations, which these authors are not the first to explore. For that reason, it becomes a bit difficult to buy when the authors claim they are the first to raise the issue of the implicit 'output-domain' knowledge missing during training, and a more complete related work section here would make the paper stronger (see for example the related work of Pan et al.)
>
> We appreciate your valuable suggestion, as it notably helps to strengthen our proposal.
> We added a paragraph and revised other paragraphs in the related work. In the second paragraph (Conflict Reduction via Knowledge Graph) in the related work section, we discuss previous works of using structured information such as knowledge graph in order to reduce input-domain focused bias.  It discusses the following references as follows.
> * Kenneth Marino, Ruslan Salakhutdinov, and Abhinav Gupta. The more you know: Using knowledge graphs for image classification. In Proceedings of the IEEE Conference on Computer Vision and Pattern Recognition, pp. 2673–2681, 2017.
> * Xiaolong Wang, Yufei Ye, and Abhinav Gupta. Zero-shot recognition via semantic embeddings and knowledge graphs. In Proceedings of the IEEE conference on computer vision and pattern recognition, pp. 6857–6866, 2018.
> * Michael Kampffmeyer, Yinbo Chen, Xiaodan Liang, Hao Wang, Yujia Zhang, and Eric P Xing. Rethinking knowledge graph propagation for zero-shot learning. In Proceedings of the IEEE/CVF conference on computer vision and pattern recognition, pp. 11487–11496, 2019.
> * Lewei Yao, Jianhua Han, Youpeng Wen, Xiaodan Liang, Dan Xu, Wei Zhang, Zhenguo Li, Chunjing Xu, and Hang Xu. Detclip: Dictionary-enriched visual-concept paralleled pre-training for open-world detection. Advances in Neural Information Processing Systems, 35:9125–9138, 2022
> * Chenchen Zhu, Fangyi Chen, Uzair Ahmed, Zhiqiang Shen, and Marios Savvides. Semantic relation reasoning for shot-stable few-shot object detection. In Proceedings of the IEEE/CVF conference on computer vision and pattern recognition, pp. 8782–8791, 2021.
>
> To the best of our knowledge, previous works focus on the effective use of external knowledge, while undescribed world knowledge over latent objects in human labeling of target training data has not been utilized and even recognized. Please check the related work section for more detailed discussions.
>
> > Weakness3. The conclusion and future work section does not really contain recommendations for future work.
>
> In the conclusion of our paper's revised edition, we implemented the following revisions:
> Beyond using the plug-in model, effectively harmonizing the undescribed world knowledge with input-domain focused bias on a simple network still remains an open question. Also, we hope that our approach will evoke further research on diverse visual domains.

---

> > ### Author Response · Authors · 2023-11-17
> > **Thanks for your comments!**
> >
> > > Question1. In A.1, you refer to Tables 7a and 7c although I believe you mean Figure 7 and 7c.
> >
> > * ‘Table 7a’ and ‘Table 7c’ is revised into ‘Figure 7a’ and ‘ Figure 7c’
> >
> > > Question2. Conclusion: the phrasing "harbors unseen knowledge from human labeling" was difficult to parse and ambiguous.
> >
> > With the redefinition of ‘ODB’, we revised the following sentence.
> > In this paper, we highlighted the conflict of input-domain focused inductive bias and undescribed world knowledge over latent objects in human labeling.”
> >
> > > Question3. Conclusion: "on vision transformer architecture" >> "on the Vision transformer architecture"
> >
> > * In the conclusion section, “on vision transformer architecture” is revised into “on Vision transformer architecture”
> >
> > > Question4. Conclusion: "in qualitative and quantitative analysis on its results" >> "through qualitative and quantitative analysis of its results"
> >
> > * In the conclusion section, "in qualitative and quantitative analysis on its results" is revised into “through qualitative and quantitative analysis of its results”
> >
> > > Question5. Table 2: Why not show the standard deviation if in effect these results were run using 5 random seeds? It seems relevant in your comparison to the baseline which is quite close (84.80 vs. 84.40). You cannot use the term 'significant' in the Ablation study section of Section 5.1 if you do not show these standard deviations.
> >
> > We have revised the results in Table 2 to include the standard deviation.
> > We also corrected a minor numerical discrepancy in Table 1, where the result of ViT B/16 on IN1K differed from Table 2, despite originating from the same experiment. Note that this modification does not impact our paper's main contention.
> >
> > > Question6. Table 2: explain what w/o Pos. stands for in table caption, even if you also say it in the text (if somebody glances quickly at the table.)
> >
> > In the caption of Table 2, we have added the following explanation: “We study the impact of visual dependency disconnection (Visual Disc.), diversity loss (Diversity), not using positional encoding (w/o Pos.), and integration module with visual features (Integration).”
> >
> > > Question7. Formatting of references generally needed, use {} around text which should be case-sensitive in the .bib-file. (e.g., "mixup" >> "MixUp" for Zhang et al.)
> >
> > We also agree with this formatting, but formally published versions of the paper use mixup without upper scaling. For this reason, we left ‘mixup’ as its original paper. To be clear, we check the missed uppercase words in references.
> >
> > > Question8. In the end of Section 5.2, it would be great to specify either in the text or in the figure caption which classes numbers 1173 and 1813 correspond to. Are they the quill and paperknife or other visually similar classes?
> >
> > The numbers in each tile correspond to the assigned latent object index to an image patch described in Section 3.2 (Method Description - Latent Object Extraction from Visual Features). 1173 and 1813 are indexes of O latent objects, and we set O(=2048) as a hyperparameter.
> > We revised the caption in Figure 6 and also revised related sentences to use one notation ‘O’ to represent the number of the latent object.
> >
> > > Question9. 5.1 title: quantiTative*
> >
> > * Section 5.1 title is revised into “Quantitative Analysis”
> >
> > > Question10. Section 4: "doesn’t" >> "does not" (too informal)
> >
> > * In Section 4, "doesn’t require" is revised into “does not require”
> >
> > > Question11. Section 4: "provide series of analysis the impact" >> "provide a series of analysEs OF the impact"
> >
> > * In Section 4, "provide series of analysis the impact" is revised into “provide series of analyses of the impact”
> >
> > > Question12. The second paragraph of the related work contains a duplicate sentence "Lemesle et al...". Should be removed.
> >
> > * The duplicated sentence is Related work section is removed
> >
> > > Question13. Fig. 3 is a nice figure. Howeve,r it currently says "L_diveristy" whereas I think you want to say "L_diversity".
> >
> > * In Figure 3, "L_diveristy" in Figure 3b is revised into "L_diversity"

---

### Official Review · Reviewer_9BG9 · 2023-11-03

**Soundness:** 2 fair
**Presentation:** 2 fair
**Contribution:** 2 fair
**Rating:** 6
**Confidence:** 3

**Summary:**

This paper analyzes the inductive bias problem in existing methods and proposes Output-Domain focused Biasing (ODB) training strategy to overcome this limitation without external resources. The authors implemented ODB on  a vision transformer architecture and achieved improvements on image classification benchmarks.

**Strengths:**

1. This paper raises the interesting issue of inductive bias in existing methods.

**Weaknesses:**

1.Humans benefit from world information being able to avoid input domain bias. This actually benefits from humans having more prior information and being able to build better semantic relationships between classes. While Output-Domain focused Biasing (ODB) is more like a feature enhancement method, which improves performance through decoupling and enhancement of features.

2.Figure 2 shows that the class centroids between some semantically unrelated classes are close to each other. Using triplet loss or contrast loss can also achieve the effect of widening the class centroids distance. It is recommended to add comparative experiments with this type of method.

3.The paper is not easy to follow, especially the descriptions of Visual Dependency Disconnection and Non-visual Feature Structuring need more details.

4.The proposed method has limited improvement in performance.

**Questions:**

1.Is the ODB method equally effective in convolutional networks?

---

> ### Author Response · Authors · 2023-11-17
> **Thanks for your comments!**
>
> Thank you for all your efforts and thoughtful comments. We have highlighted the revisions in our paper with brown text color based on the recommendation.
>
> > Weakness1. Humans benefit from world information being able to avoid input domain bias. This actually benefits from humans having more prior information and being able to build better semantic relationships between classes. While Output-Domain focused Biasing (ODB) is more like a feature enhancement method, which improves performance through decoupling and enhancement of features.
>
> We totally agree with the benefit of human knowledge. However, we hold a differing view that the absence of human knowledge is a weakness. because this paper aimed to improve neural networks and actually intended not to use external resources. This is one of the common goals of AI literature to reduce human efforts in building better AI for the last decades.
>
> > Weakness2. Figure 2 shows that the class centroids between some semantically unrelated classes are close to each other. Using triplet loss or contrast loss can also achieve the effect of widening the class centroids distance. It is recommended to add comparative experiments with this type of method.
>
> Thanks for your suggestions.
>
> We would like to emphasize that decoupling is one of the phenomena induced by ODB(revised to LLB). The primary distinction between ODB and traditional contrastive learning resides in the suppression of cluster positioning based on similarity within the input domain. This effect does not occur in methods simply oriented to decoupling representations.
> For example, final representations generated from visually similar objects should be located in adjacent clusters even if the clusters are decoupled as shown in Figure 4. (In (c),  class 0-1 are very close, even overlapped, compared to 0-2. However, in (d), class 2 completely separates class 0 and class 1.)
> For this reason, ODB more aptly aligns with a method aimed at inducing a bias relatively insensitive to input-domain features, as opposed to a method solely focused on decoupling.
>
> As the reviewer’s recommendation, we added the qualitative analysis of models using contrastive learning on the same data in Section 3.1. Figure 11d shows that the visually similar cluster centroids are not detached as ODB does.
> The results are briefly shown as follows.
> Using contrastive learning, the centroids are slightly decoupled compared to supervised learning. However, contrastive learning still fails to widen the gap between ’462: Broom’ and ’746: Puck’, where two class labels are visually similar in stick parts, but semantically distinguished between brush and puck. This observation shows that contrastive learning on CNN can reduce the input-domain focused bias, but still fails in some cases.
> Please check Section C.2 in Appendix for figures and more details.
>
> Plus, the reason why we do not consider contrastive learning is that SOTA models on ViT do not include explicit loss based contrastive learning.
>
> > Weakness3. The paper is not easy to follow, especially the descriptions of Visual Dependency Disconnection and Non-visual Feature Structuring need more details.
>
> To improve the readability, we revised Section 3.2 including  ‘Visually Dependency Disconnection’ and ‘Non-visual Feature Structuring’  paragraphs.
>
> The revisions are briefly shown as follows.
>
> 1) Notation definition of visual and non-visual features
> We revised the notation for ‘visual features’ in Section 3.1 and ‘non-visual features’ in Section 3.2.
>
> 2) Section 3.2 Visually Dependency Disconnection
> For clear readability, we defined a notation for the output of the disconnect network as patch-wise non-visual features in the third paragraph of Section 3.2. Also, we added explanations of each notation in Equation (3).
>
> 3) Emphasized points
> To improve the clarity, we generally clarified the emphasized points in Section 3.2. We updated details at main components in our implementation.
>
> Please check Section 3.1 and Section 3.2 for more details.
>
> > Weakness4. The proposed method has limited improvement in performance.
>
> Our results show consistent and statistically significant improvement. We hope that the contribution on raising a new problem and its generality are more considered rather than the significance of performance gain determined by an unclear threshold.

---

> > ### Comment · Reviewer_9BG9 · 2023-11-23
> >
> > Thanks for authors' efforts to address my concerns. I raise my score from 5 to 6, and I hope authors can add these additional atoms into final version.

---

> ### Author Response · Authors · 2023-11-17
> **Thanks for your comments!**
>
> > Question1. Is the ODB method equally effective in convolutional networks?
>
> The effect of the ODB (revised to LLB) method is basically not relying on specific architecture if its representations are directly affected by loss, so we think it is also effective in convolutional networks. The only reason why we selected ViT is because of its wide use and state-of-the-art performance.
> To provide more insights, as in Section 3.1 preliminaries, we visualized the distribution of centroids of all output features in each class of ImageNet, extracted from ResNet50 as shown in Section C.1  in Appendix. We found that adjacent classes from Figure 2b, which are visually similar but semantically unrelated, were also closely positioned in CNN (Broom, Puck, Crutch, Mop) (Figure 11c) . We think that the dominance of the input-domain focused bias also exists in CNNs, and therefore ODB can be effective in CNNs. Check Section C.1 in Appendix for more details.

---

### Author Response · Authors · 2023-11-17
**Thanks for your comment!**

Thank you for all your efforts and valuable comments. We responded and revised our papers to reduce the concerns.

We note that important terms are changed as follows, accepting the recommendation of a reviewer not to use  “output-domain” term.

1) ‘output-domain focused inductive bias’ -> label-focused inductive bias
2) implicit relations over latent objects used for  human labeling -> undescribed world knowledge
3) ‘Output-domain focused inductive biasing (ODB)’ (the method) -> Label-focused Latent-object Biasing (LLB)’
* Label-focused inductive bias is inductive bias determined by categorization of latent objects based on only labels, while undescribed world knowledge indicates all unexpressed world knowledge on the objects in human labeling.

Please note the change of sentences related to the terms (purple-colored texts) while checking the other concerns.

---

### Comment · Reviewer_CuBT · 2023-11-22
**Maintain my accept score (8)**

Thank you to the authors for your thorough replies in this rebuttal process. The paper has only improved in clarity since the submission, and I maintain my score at 8. I don't see the weaknesses listed by my fellow reviewers as strong enough reasons for rejection.

---

### Meta-Review · Area_Chair_LrR4 · 2023-12-06

**Metareview:**

The paper in discussion presents a novel method called Label-focused Latent-object Biasing (LLB), previously referred to as Output-Domain focused Biasing (ODB), aimed at addressing the issue of inductive bias in vision neural networks. The authors propose LLB to enhance the classification performance by emphasizing inductive biases based on output labels, which helps avoid the dominance of input-domain-focused bias. This method involves learning intermediate latent object features, decoupling visual dependencies, capturing structured features optimized for classification, and integrating these features for prediction.

Strengths
------------
Reviewer feedback indicates that the paper is well-structured and written, with clear improvements in clarity following revisions. The reviewers acknowledge the novelty of the approach and its thorough inspection through both quantitative and qualitative analyses. They appreciate the efforts made by the authors to address the concerns raised, such as redefining certain terminologies for better clarity and adding comparative experiments as suggested.

Weaknesses
------------
However, reviewers also point out some weaknesses. For instance, there are concerns about the experimental improvements over baselines being modest and the method's potential limitations in terms of performance gains. Moreover, some reviewers suggest that the paper could benefit from a more detailed related work section and clearer future work recommendations.

Despite these concerns, the overall feedback from the reviewers is positive, with scores indicating a lean towards acceptance. The authors' proactive engagement in addressing reviewer comments and revising the paper accordingly seems to have positively influenced the reviewers' opinions.

**Justification For Why Not Higher Score:**

The paper is indeed a valuable contribution and the AC recommends its acceptance. The paper may benefit from further revision to improve the presentation quality of the paper and find experimental setups where gains can be more significant.

**Justification For Why Not Lower Score:**

The paper has three accepted scores. No basis to overturn the reviews and reject the paper.

---

### Decision · Program_Chairs · 2024-01-16

Accept (poster)